# Contrast-Associated Acute Kidney Injury Requiring Continuous Renal Replacement Therapy in A Neonate with Aortic Stenosis—A Case Report

Catalin Cirstoveanu [1,2], Carmina Georgescu [2,*], Nicolae Ruxandra [2], Mihaela Bizubac [1,2,*], Eliza Cinteza [3,4], Corina Maria Vasile [4,5], Cristina Filip [3,4] and Irina Margarint [6,7]

1 Department of Neonatal Intensive Care, "Carol Davila" University of Medicine and Pharmacy, 020021 Bucharest, Romania
2 Neonatal Intensive Care Unit, M.S. Curie Children's Hospital, 041451 Bucharest, Romania
3 Department of Pediatrics, "Carol Davila" University of Medicine and Pharmacy, 020021 Bucharest, Romania
4 Pediatric Cardiology, M.S. Curie Children's Hospital, 041451 Bucharest, Romania
5 Department of Pediatric and Adult Congenital Cardiology, University Hospital of Bordeaux, 33600 Pessac, France
6 PhD School Departemnt "Carol Davila", University of Medicine and Pharmacy, 020021 Bucharest, Romania
7 Pediatric Cardiovascular Surgery, M.S. Curie Children's Hospital, 041451 Bucharest, Romania
* Correspondence: carmina.georgescu@yahoo.com (C.G.); mihaela.bizubac@drd.umfcd.ro or anamihaeladogaru@yahoo.com (M.B.)

**Abstract:** Background: Acute kidney injury occurs commonly in the Neonatal Intensive Care Unit and is associated with increased mortality and morbidity. We report a case of a neonate with congenital heart disease who developed acute kidney injury after cardiac surgery, administration of iodinated contrast media for cardiac catheterization, and a combination of nephrotoxic drugs. Case report: A term neonate without a prenatal diagnosis of congenital heart disease and with a good postnatal transition was transferred at 13 days of life to the MS Curie Emergency Hospital for Children, Newborn Intensive Care Unit, from a regional hospital where he was admitted at 10 days of life with severe general status, respiratory distress, cyanosis, and arterial hypotension. The cardiac ultrasound detected critical aortic valve stenosis, hypoplastic descending aorta, acute heart failure, and pulmonary hypertension. The patient was intubated and mechanically ventilated and received antibiotherapy (meropenem, vancomycin, and colistin), inotropic and vasoactive support (epinephrine, norepinephrine, dopamine, and milrinone), and diuretic support (furosemide, aminophylline, and ethacrynic acid). A balloon aortic valvuloplasty was performed several hours after admission, but after two days the patient required reintervention by open heart surgery due to relapsing severe aortic stenosis. He developed oligo-anuria, generalized edema, and altered renal function tests on the second postoperative and fourth day post-contrast media administration. Continuous renal replacement therapy was initiated for 75 h, leading to almost instant improvement in blood pressure, then diuresis and creatinine levels. The patient required long-term treatment for heart, respiratory, and liver failure. He was discharged at almost four months of age with normal renal function tests, blood pressure, and good urine output without diuretic support. The literature review indicates that contrast-associated acute kidney injury (CA-AKI) requiring continuous renal replacement therapy is rare. Conclusions: Our current case proves that iodinated contrast media administration in a neonate with concomitant insults, such as cardiac surgery for a specific pathology, aortic stenosis, coarctation, arch stenosis, arterial hypotension, and administration of nephrotoxic drugs, may lead to severe kidney injury.

**Keywords:** contrast-associated acute kidney injury; continuous renal replacement therapy; congenital heart disease; aortic stenosis; cardiac surgery; nephrotoxic medication; neonatology

## 1. Introduction

Acute kidney injury (AKI) is common in the Neonatal Intensive Care Unit (NICU), particularly in high-risk patients and especially those with congenital heart disease undergoing cardiac surgery and those receiving nephrotoxic drugs and iodinated media contrast. According to the largest international study, the incidence of AKI was 37% in term neonates admitted to the NICU [1]. AKI was independently associated with increased mortality (9.7% vs. 1.4%) and prolonged hospital stay. The incidence of neonatal AKI was even higher among specific sub-populations at high risk.

Nephropathy induced by iodinated media contrast administration is considered a major risk factor for developing acute kidney injury, associated with increased hospitalization, costs, irreversible kidney injury, need for renal replacement therapy, and mortality [2]. Contrast nephrotoxicity usually develops with concomitant insults, such as arterial hypotension, administration of nephrotoxic drugs, and underlying comorbidities that may compromise renal blood flow. Contrast-associated AKI (CA-AKI) incidence in the pediatric population ranges from 3.3% to 10%, depending on the patient cohort [3,4].

Acute kidney injury occurs in almost one in two neonates experiencing cardiac surgery, leading to a longer duration of mechanical ventilation and hospital stay [5]. The mortality rate of neonates requiring dialysis after cardiac surgery is 6.4 times higher [5]. Recently, the Neonatal and Pediatric Heart and Renal Outcomes in Newborns (NEPHRON) collaborative found an overall incidence of AKI in 53.8% in neonates undergoing congenital heart surgery, varying from 27% to 86% between institutions [6]. The peak in the prevalence of AKI was on the first postoperative day. However, only stage 3 AKI was associated with increased mortality.

Hemodynamic fluctuations caused by cardiopulmonary bypass and cross-clamping of the aorta, perioperative low cardiac output syndrome, vasoactive drugs, albumin administration, transfusions, inflammatory factors, and clotting disorders are involved in the development of AKI after cardiac surgery [7].

Critically ill neonates are frequently exposed to nephrotoxic drugs. A recent study evaluating the impact of combinations of nephrotoxic drugs in a multicenter database, including 8286 neonates exposed to nephrotoxic drugs, found an incidence of AKI of 17% [8]. Moreover, the increased duration of drug exposure was associated with an increased risk of acute kidney injury.

## 2. Case Report

We report the case of a male newborn from an investigated high-risk pregnancy. The birth was natural, with a birth weight of 2980 g and an Apgar score of 9 and 10 at 1 and 5 min, respectively, with a good postnatal transition. He was readmitted to the hospital at ten days of life with severe general status. The echocardiography revealed critical stenosis of the aortic valve, severe systolic biventricular dysfunction, and pulmonary hypertension (Figure 1). Prostaglandin E1 was initiated in high-dosage continuous infusion (0.15 mcg/kg/min). He was transferred to the Neonatal Intensive Care Unit of the MS Curie Children's Hospital at 13 days of age with a severe general condition and hemodynamic instability. The patient was immediately intubated and mechanically ventilated due to respiratory distress. Intravenous central venous lines were placed for nutrition and medication administration. Central arterial lines were inserted for continuous invasive blood pressure monitoring and femoral artery cannulation for cardiac catheterization.

After five hours of admission, we performed diagnostic cardiac catheterization using 2 mL/kg iohexol 300 for iodinated contrast, which was subsequently followed by balloon aortic valvuloplasty (4.3 mm aortic ring, Tyshak II balloon size 4/20 mm, inflated to nominal pressure 4. 5 atm, twice), which improved hemodynamic parameters (preprocedural femoral blood pressure 17/10/12 mmHg/postprocedural left ventricular pressure 74/19/24 mmHg, ascending aortic pressure 68/40/50 mmHg). Post-procedure, the patient experienced mild hypotension, metabolic acidosis, and decreased urine output despite inotropic-vasoactive support (epinephrine and dopamine) and diuretic support

(furosemide, and aminophylline). The next day, cardiac ultrasound showed a unicuspid aortic valve with a severely restricted opening, an abdominal aortic velocity of 0.2 m/s, severe biventricular systolic dysfunction, and severe pulmonary hypertension.

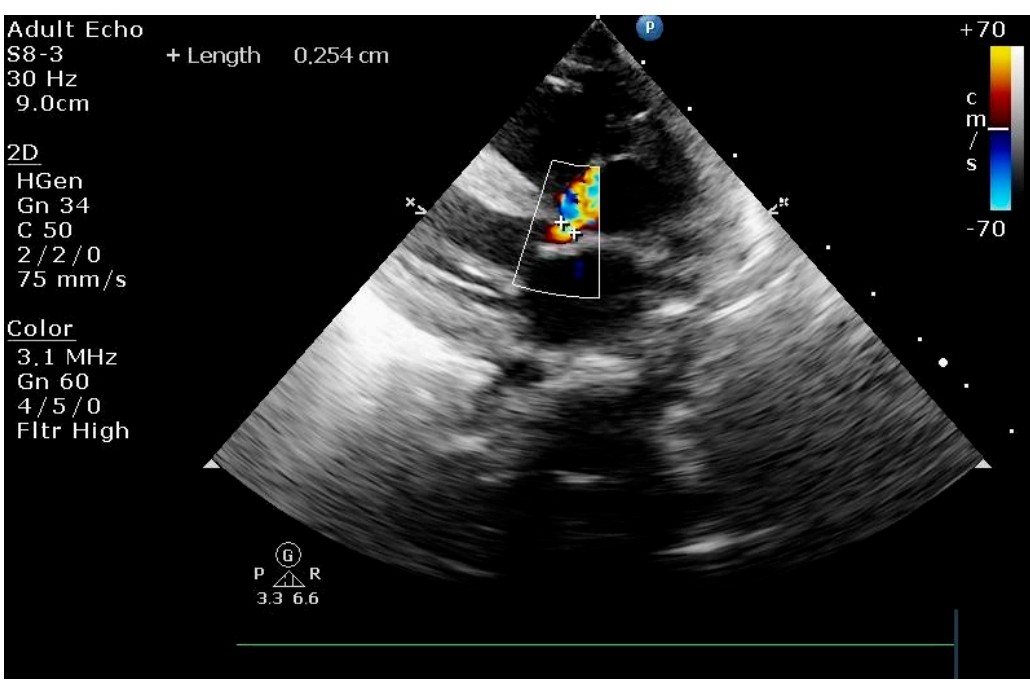

**Figure 1.** Transthoracic echocardiography performed in PLAX view after balloon aortic valvuloplasty: residual aortic valve stenosis.

The following day after the aortic balloon valvuloplasty, the multidisciplinary team decided that a reintervention through open heart surgery was necessary for the residual severe aortic valvular stenosis. The surgery lasted 5 h (55 min of cardiopulmonary bypass and 40 min of aortic clamping). After the procedure, the patient remained hemodynamically unstable with low cardiac output syndrome despite increased vasoactive-inotropic support and presented heavy active bleeding on the chest tubes, requiring several transfusions with blood products and tranexamic acid administration. Postoperative echocardiography revealed moderate residual aortic valvular stenosis, severe biventricular systolic dysfunction, and pulmonary hypertension.

On the second postoperative day (day four after cardiac catheterization), the patient became oligo-anuric despite increased diuretic support in continuous infusion (furosemide, aminophylline, and ethacrynic acid), with generalized edema and significant positive fluid balance. Serial renal function tests revealed a rising trend in creatinine and urea levels. As the patient presented stage 3 AKI according to the neonatal KDIGO (Kidney Disease Improving Global Outcomes) classification, continuous renal replacement therapy was initiated the following day, and diuretic medication was discontinued. Vascular access for the CRRT procedure was obtained through a double-lumen 6Fr catheter inserted by the Seldinger technique at the patient's bedside under ultrasound guidance into the right femoral vein. Anticoagulation therapy for the circuit was initiated with heparin in a continuous infusion, and the activated clotting time was maintained between 180 and 200 s. Continuous veno-venous hemodiafiltration was well tolerated, and the patient remained hemodynamically stable, allowing the increase in the fluid removal and the decrease in the vasoactive support. There was a serial fall in creatinine levels, and urine output was re-established after 4 h. Continuous renal replacement therapy (CRRT) was maintained for 75 h. Sternal closure was performed on site in the NICU on postoperative day 5, while the patient was on CRRT, without major hemodynamic impact.

Several complications developed during hospitalization have been reported for the patient despite solving AKI: the prolonged need for inotropic and vasoactive support (77 days), initial extubating failure and prolonged need for invasive mechanical ventilation (71 days), pulmonary edema due to sustained mitral regurgitation, and three episodes of pulmonary hemorrhage due to coagulopathy and high capillary hydrostatic pressure. In addition, he developed acute liver failure after heart failure exacerbation two weeks after surgery. Persistent conjugated hyperbilirubinemia recovered after two months of treatment with ursodeoxycholic acid, alpha lipoic acid, phenobarbital, cholestyramine and serial albumin administration, vitamin K, and fresh frozen plasma. The day after cardiac surgery, a 2-grade intraventricular hemorrhage was noted, but without any apparent clinical significance. He was discharged from the hospital at 3 months and 25 days with mild residual aortic valve stenosis and grade 2 aortic regurgitation. At hospital discharge, renal function tests and blood pressure were normal for age.

## 3. Discussions

Current diagnostic criteria for neonatal acute kidney injury are mainly based on serum creatinine level and hourly urine output but are not sufficiently sensitive for a prompt diagnosis. The early phase of AKI is frequently missed because of the slow rise in serum creatinine level and because most neonates present non-oliguric acute kidney injury, as was our case. After the cardiac catheterization, the patient first developed polyuric acute kidney injury, followed by a decrease in urine output and a steady increase in creatinine level. The fractional excretion of urea was consistent with prerenal failure caused by hypoperfusion in the context of congenital heart disease and cardiac surgery with low cardiac output syndrome, nephropathy induced by iodinated media contrast administration, nephrotoxic medications (amikacin, colistin, and furosemide), and vasoactive agents (epinephrine, norepinephrine, and dopamine). Although the patient's renal function deteriorated after contrast media administration, which led us to suspect CA-AKI, the fundamental role of iodinated contrast media remains unclear because of the cumulative effect of multiple insults to the kidney.

Congenital heart disease is a potential cause of nephropathy due to hypotension and hypoxemia, leading to reduced renal blood flow and changes in intraglomerular hemodynamics. Hypoperfusion, objectified by a higher postoperative lactate level, is an independent risk factor for AKI [9]. Our patient had a steady high lactate level from admission until after cardiac surgery, suggesting chronic hypoperfusion of the tissues (Figure 2).

In the NEPHRON study, perioperative characteristics associated with AKI after cardiac surgery included prostaglandin infusion, longer cardiopulmonary bypass, and aortic cross-clamp duration, Society of Thoracic Surgeons-European Association for Cardio-Thoracic Surgery (STAT) score, and urine output in the operating room [6]. Several of these characteristics associated with AKI risk were also present in our patient: preoperative administration of PGE1 in high dosage, cardiopulmonary bypass, aortic cross-clamp, STAT score of 0.4, and STAT mortality category 2. However, sternal closure was delayed for five days until hemodynamic stability was achieved, and enteral feedings were initiated preoperatively, which are associated with lower AKI risk.

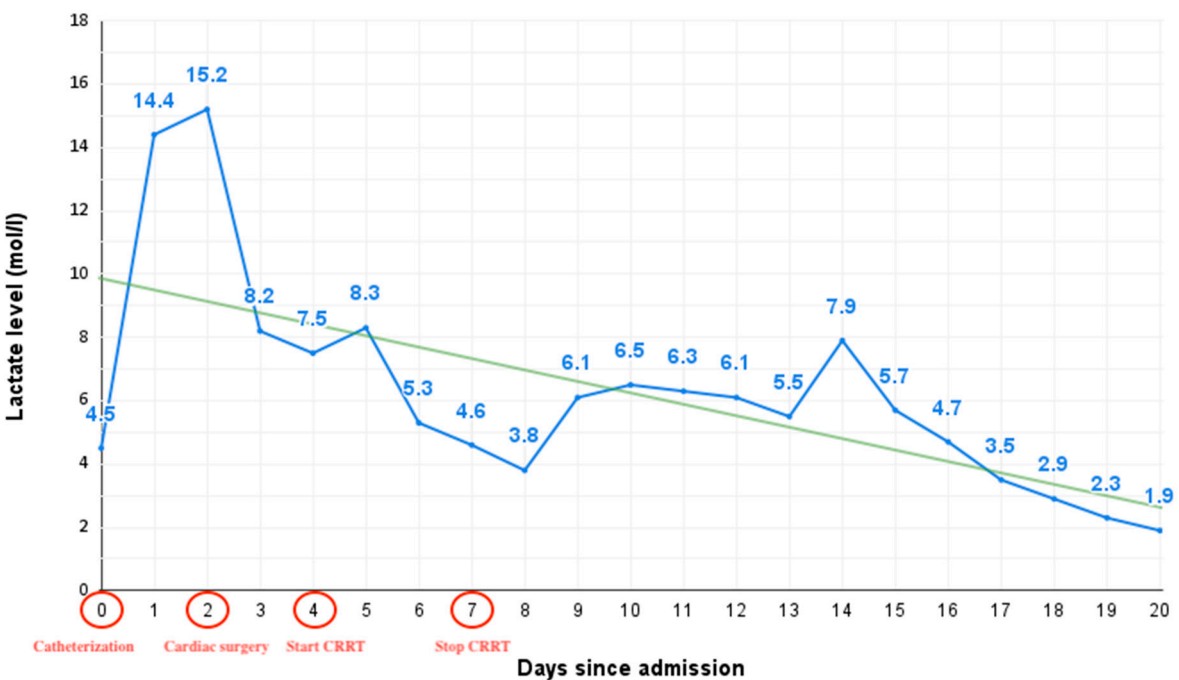

**Figure 2.** Lactate level trend from admission until the achievement of optimal tissue perfusion, marking the milestones in the timeline: day 0—catheterization, day 2—cardiac surgery, day 4—initiation of continuous renal replacement therapy (CRRT), and day 7—discontinuation of continuous renal replacement therapy.

The development of cardiac-surgery-associated acute kidney injury (CS-AKI) risk stratification tools can be useful in assessing neonates at high risk for severe AKI. An example of one such model is the vasoactive-ventilation-renal score (VVR), which provides a combined evaluation of cardiac, respiratory, and renal functions. The VVR score proved more useful than the vasoactive-inotropic score (VIS) and lactate level in predicting clinical outcomes in the early postoperative period [10]. However, the newborn population has uncertain data about postoperative timing and cut-off values. Postoperative 48th-hour VVR score (cut-off value of 22.5) was correlated with prolonged mechanical ventilation duration and hospital stay in children below one year [11]. In a multicenter study involving only newborns, the VVR score 12 h after surgery (cut-off value of 35) was superior in determining a prolonged need for mechanical ventilation [12]. A recent single-center study showed that neonates with a peak VVR ≥ 46.5 within the first 72 postoperative hours should be considered at increased risk for poor prognosis [13]. In our case, the 12 h postoperative VVR score was 90, 2.5 times higher than the cut-off value, as well as the 48 h postoperative VVR score of 95, 4 times the cut-off value, and the 72 h postoperative VRR score of 87.8, almost the double of the cut-off value (Figure 3). All these high VVR scores were suggestive of poor outcomes.

Contrast-associated acute kidney injury is defined as the development of renal failure following contrast media administration without an alternative etiology [14]. According to the Kidney Disease Improving Global Outcomes (KDIGO) definition, CA-AKI is an increase in serum creatinine by 0.3 mg/dL within 48 h after contrast media exposure or an increase by 50% within seven days [15]. According to these criteria, our patient presented CA-AKI due to an increased creatinine level by 50% in 24 h after contrast administration, before the cardiac surgery.

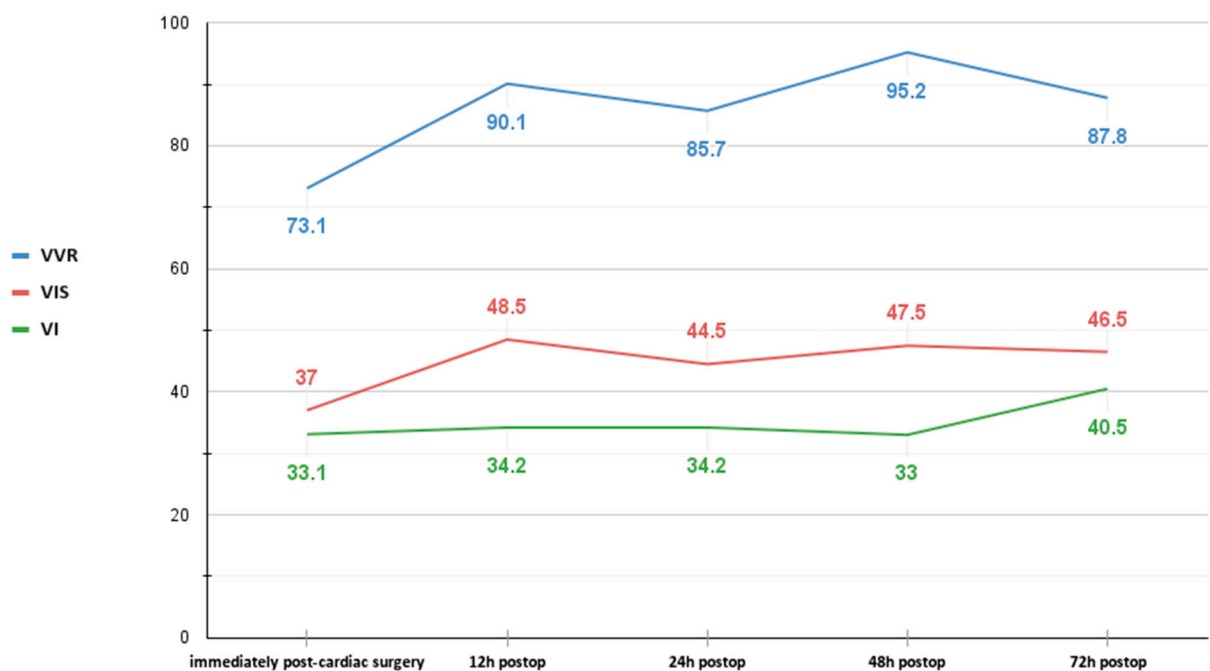

**Figure 3.** Vasoactive-ventilation-renal score (VVR), vasoactive-inotropic score (VIS), and ventilation index (VI) trend in our patient during the first 72 h post-cardiac surgery.

In the case of contrast-induced nephropathy, serum creatinine typically rises within the first 48 h after exposure, peaks at 3–5 days, and returns near to baseline within 1–3 weeks [16]. Nonetheless, irreversible renal function damage occurs in rare cases. In our case, the rise in creatinine level began after 24 h post-contrast administration and peaked after four days, when we decided to initiate CRRT, decreasing afterward (Figure 4).

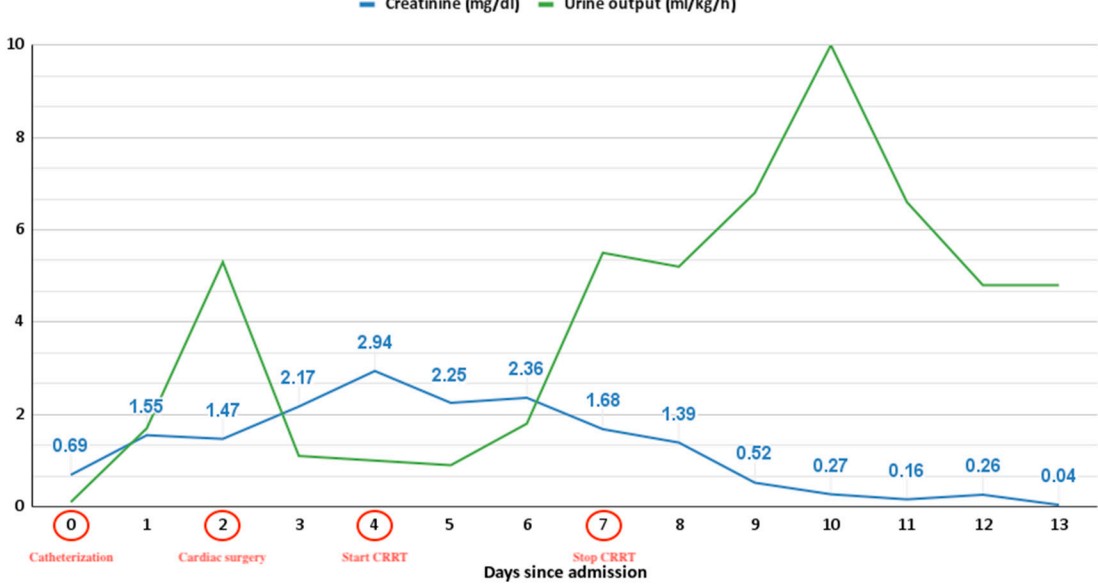

**Figure 4.** The trend in serum creatinine and urine output in our case for the first two weeks since admission, marking the major events: day 0—catheterization, day 2—cardiac surgery, day 4—initiation of continuous renal replacement therapy (CRRT), and day 7—discontinuation of continuous renal replacement therapy.

The pathophysiologic mechanism of contrast-induced nephropathy is complex. Contrast media induces the release of endothelin and adenosine, the influx of calcium ions, and the impairment of nitric oxide and prostaglandins' production, triggering renal vasoconstriction and hypoperfusion. This leads to hypoxia of the outer medulla and tubular cells [17]. Contrast media also has direct toxic effects on the tubular cells and its viscosity decreases blood oxygen delivery at the tubular cell level. These mechanisms trigger the release of reactive oxygen species, increasing oxidative stress [18]. In addition to direct contrast media effects, there are several other periprocedural factors, such as hypotension and bleeding, causing ischemic kidney injury.

Risk factors for contrast-induced nephropathy can be divided into patient-related risk factors (pre-existence of renal injury, arterial hypotension, low cardiac output, anemia, and concomitant use of nephrotoxic drugs) and procedure-related risk factors (total amount and type of iodinated contrast media, arterial versus venous route of administration, and the period between two contrast media administrations) [19–21]. Some of these risk factors were present in our patient.

The amount of contrast media administered intra-arterially is frequently considered an independent predictor for the development of nephropathy. According to the American College of Radiology, very small volumes of contrast agents should be administered in neonates and infants, typically 1.5–2 mL/kg [22]. An "ALARA" concept (as low as reasonably achievable) is usually recommended in interventional cardiology regarding radiation exposure, but also contrast agents [23]. The volume of each contrast media injection is more important than the total dosage administered. According to the drug leaflet, the recommended single injection volume of iohexol 300 for angiographic procedures in children is 1.75 mL/kg, ranging from 1.5 mL/kg to 2.0 mL/kg. When multiple injections are given, the dose should not exceed 5 mL/kg of iohexol 300. One injection of a 2 mL/kg iohexol 300 solution diluted with saline was administered to the patient to distinguish the level of the aortic valve and the origins of the coronary arteries, to visualize the opening of the valve and to prevent entering the coronary arteries. The volume of contrast media was not above the media, but CA-AKI may develop even at usual dosages when other risk factors are present.

A study reported that high contrast media load used during the preoperative angiographic examination and low postoperative cardiac output are risk factors for AKI following surgical repair in children below two years old [24]. Nonetheless, another study concluded that contrast media administration within 48 h before the cardiopulmonary bypass is not an additional risk factor for developing AKI in patients with congenital heart disease [25]. This result remains questionable in certain at-risk infants, like our patient, where several insults affecting the kidneys over a short period may compromise renal function.

A multimodal approach for renoprotection has been advocated for CA-AKI prevention [26]. Physicians should identify at-risk patients, discontinue, if possible, any potentially nephrotoxic medications at least 48 h before exposure to contrast media, minimize contrast volume administered, and target euvolemia.

There are no evidence-based guidelines for preventing and treating contrast-media-associated AKI in children; most prevention and management strategies follow adult recommendations. Hydration before and after the procedure is the most effective measure to reduce the occurrence of contrast-induced nephropathy. Fluid administration increases urine flow, reduces the concentration of contrast medium in the tubules, and promotes the clearance of contrast medium, thereby limiting the time of exposure of tubular cells to contrast medium [23]. The optimal type, amount, timing, and duration of fluid administration are still under investigation. Intravenous administration of normal isotonic saline is the most used.

Currently, no available drug is approved for the prevention of contrast-associated AKI. In large-scale randomized trials, N-acetylcysteine, aminophylline, and other drugs, such as dopamine in "low-dose," failed to confirm any beneficial preventive effect.

In the case of failure to prevent AK, the development of fluid overload > 10% of the body weight, and the inability to maintain input–output fluid balance despite optimal medical treatment, renal replacement therapy should be initiated [27].

One session of intermittent hemodialysis can efficiently remove 60% to 90% of contrast media from the blood [28]. However, continuous renal replacement therapy (CRRT) is preferred in hemodynamically unstable infants due to slower fluid removal, higher mean arterial pressure at the end of the treatment, and lower need for increasing vasopressor support [29].

The initiation of CRRT in neonates is often delayed because of several technical difficulties related to vascular access and lack of experience. However, recent advances have made CRRT increasingly accessible and successful in neonates. Approximately 10% of neonates with AKI require renal replacement therapy, depending on the AKI etiology [30]. From 2015 until 2021, we performed 44 renal replacement therapies in our unit, and 56.8% of the procedures were in patients who had undergone cardiac surgery.

A previous study in our unit that analyzed 28 patients who underwent CRRT showed a positive correlation between increased VIS score and CRRT, with statistical significance [31]. VIS > 48 was established as the tie-breaking criterion for CRRT, with a sensitivity of 62.1% and a specificity of 90.6% ($p < 0.0001$). In the case presented, VIS at 12 h post-cardiac surgery was 48.5, and VIS at 48 h was 47.5, near the cut-off criterion for CRRT.

Arterial hypotension is the most frequent complication of CRRT, present in up to 30% of cases [32]. In our previous study, two-thirds of the patients undergoing CRRT developed mild arterial hypotension requiring additional vasopressor support. However, CRRT was very well tolerated in this case (Figure 5), with a steady increase in mean arterial blood pressure which allowed the gradual increase in fluid removal from 5 mL/h to 30 mL/h in the first 12 h and the decrease in inotropic-vasoactive support (VIS decreased from 46.5 pre-CRRT to 16.5 post-CRRT).

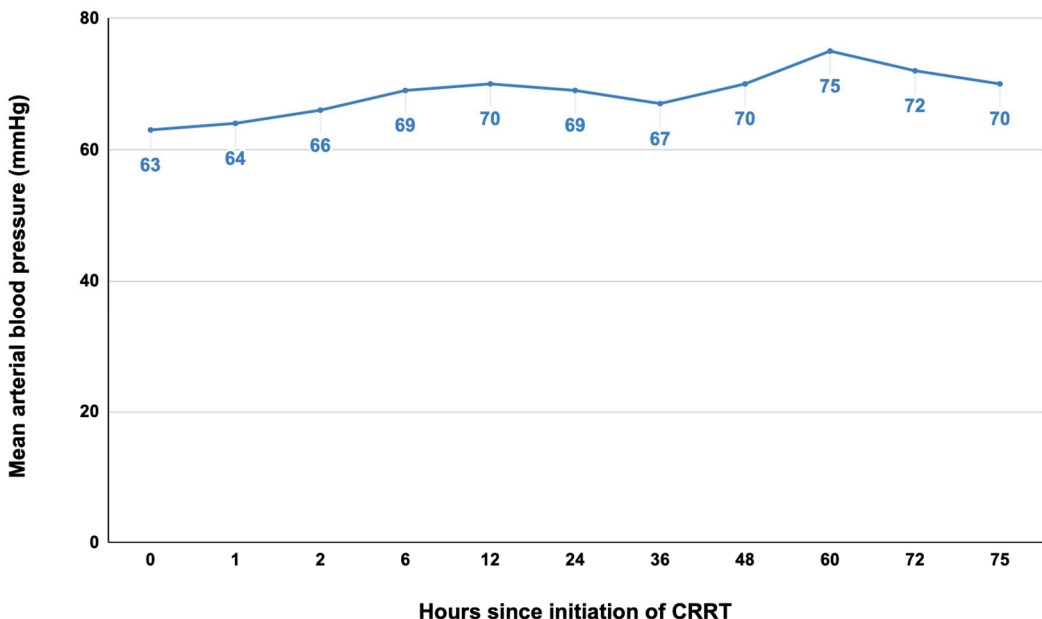

**Figure 5.** The trend in mean arterial blood pressure during the 75 h of continuous renal replacement therapy.

After 75 h, CRRT was discontinued due to circuit clotting when we decided against continuing with the therapy because the patient had already resumed diuresis, and fluid overload went into remission. Based on our experience, the short duration of CRRT required in this case pleads for CA-AKI exacerbated by multiple associated renal insults (cardiac surgery and nephrotoxic and vasoactive drugs).

## 4. Conclusions

AKI remains challenging in the neonatal period, frequently appearing due to multiple insults. Since diagnostic and therapeutic interventional cardiac procedures are becoming increasingly frequent, contrast-associated AKI has become a topic of increasing interest to physicians. The present investigation illustrates that even normal-dose contrast administration can lead to acute renal failure in a cardiac surgery patient with low cardiac output syndrome.

Identifying neonates at risk of developing AKI after contrast media administration or cardiopulmonary bypass allows for early detection and the ability to adopt preventive strategies, such as dose adjustments of nephrotoxic drugs and maintaining hydration euvolemia. Once AKI is established, there is no specific treatment. Renal replacement therapy initiated in a timely manner is the only proven effective treatment for advanced-stage AKI. Based on our experience, CRRT is a safe and effective procedure in CA-AKI; therefore, it should be considered in neonates not responding to conventional medical strategies.

**Author Contributions:** Conceptualization, C.C. and I.M.; methodology, N.R. and M.B.; software, C.G.; validation, C.C., E.C. and I.M.; formal analysis, C.F. and C.C.; investigation, M.B.; resources, C.C., C.F., E.C. and I.M.; data curation, C.G.; writing—original draft preparation, C.G., C.C. and M.B.; writing—review and editing, C.M.V.; visualization, C.C.; supervision, C.C., E.C. and C.F.; project administration, I.M. All authors have read and agreed to the published version of the manuscript.

**Funding:** This research received no external funding.

**Institutional Review Board Statement:** The study was conducted in accordance with the Declaration of Helsinki and approved by the Institutional Review Board of MS Curie Children's Hospital (approval no. 11056/15.03.2023).

**Informed Consent Statement:** Informed consent was obtained from all subjects involved in the study. Written informed consent has been obtained from the patient to publish this paper.

**Data Availability Statement:** Not applicable.

**Conflicts of Interest:** The authors declare no conflict of interest.

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
