# Peer review of "Contrast-Associated Acute Kidney Injury Requiring Continuous Renal Replacement Therapy in A Neonate with Aortic Stenosis—A Case Report"

_pediatrrep, doi:10.3390/pediatric15020028_

Round 1

Reviewer 1 Report

1. "... was transferred at 13 days of life at to “M.S. Curie” Emergency Hospital ..." Kindly replace the word "at" to "to". 

2. On line 37, I'd prefer the authors use the term "altered" renal function tests instead of "increased" renal function tests. It's unclear. 

3. Line 91. "The patient was immediately intubated and mechanically ventilated due to respiratory distress.(period) Intravenous central venous lines were...." Kindly add a period. 

4. On line 143, "A grade 2 intraventricular hemorrhage later complex cardiac interventions observed the day after cardiac surgery without apparent clinical significance. ". Kindly rephrase.

5. The authors have referred to the contrast-related AKI as either "CA-AKI" and "CI-AKI" throughout the manuscript. A unique term—either of the two—would be preferred. Note: I  have assumed CI-AKI is what they mean by contrast-induced AKI. The expanded version of CI-AKI has not been mentioned.

Overall, the report is unique and is written well. There are minor grammar and punctuation errors, for instance, on Line 267, besides those mentioned above. Given the content, the authors have described the case very well. 

I'd recommend the authors describe the timeline a bit more clearly in the abstract. Besides that, the authors have done a commendable job in describing the case in their report.

The language is quite good. I believe a few typographic and punctuation errors have found their way inadvertently during reviewing. 

Author Response

Dear Reviewer,

We appreciate that you took the time to review our manuscript.

  1. "... was transferred at 13 days of life atto“M.S. Curie” Emergency Hospital ..." Kindly replace the word "at" to "to". 

We modified accordingly

  1. On line 37, I'd prefer the authors use the term "altered" renal function tests instead of "increased" renal function tests. It's unclear. 

We modified accordingly.

  1. Line 91. "The patient was immediately intubated and mechanically ventilated due to respiratory distress.(period)Intravenous central venous lines were...." Kindly add a period.

We modified accordingly

  1. On line 143, "A grade 2 intraventricular hemorrhage later complex cardiac interventions observed the day after cardiac surgery without apparent clinical significance. ". Kindly rephrase.

We rephrased accordingly

  1. The authors have referred to the contrast-related AKI as either "CA-AKI" and "CI-AKI" throughout the manuscript. A unique term—either of the two—would be preferred. Note: I  have assumed CI-AKI is what they mean by contrast-induced AKI. The expanded version of CI-AKI has not been mentioned.

We modified accordingly and spelled the abbreviation after the first mentioning in the manuscript.

Overall, the report is unique and is written well. There are minor grammar and punctuation errors, for instance, on Line 267, besides those mentioned above. Given the content, the authors have described the case very well. 

We modified accordingly.

I'd recommend the authors describe the timeline a bit more clearly in the abstract. Besides that, the authors have done a commendable job in describing the case in their report.

Kind regards,

The authors

Reviewer 2 Report

This report presents a neonate with congenital heart disease who developed AKI after undergoing cardiac surgery, receiving contrast media, and being exposed to nephrotoxic drugs. The condition of this patient improved significantly after undergoing CRRT, leading to improved renal function, blood pressure, diuresis, and lactate levels. The rarity of contrast-associated AKI requiring CRRT in neonates is highlighted, along with the importance of early detection and preventive strategies for high-risk neonates, especially when multiple risk factors are present. The study concludes that CRRT is a safe and effective treatment for advanced-stage AKI in neonates. Overall, this manuscript is well-organized and thoughtfully written, providing valuable insights for managing AKI in neonates and emphasizing the need for cautious administration of contrast media and nephrotoxic drugs in this population.

Author Response

Dear Reviewer,

We appreciate your taking the time to review our manuscript and the kind word regarding our work.

Kind regards,

The authors

Reviewer 3 Report

This case report is related to contrast associated acute kidney injury requiring CRRT in a neonate with aortic stenosis. This is very extensive and detailed report based on authors own experience with very good discussion and clear recommandation in conclusion.

I have only some technical comments.

Mark Figure 1 in the text.

Abbreviation KDGIO and CRRT should be named fully when used for the first time in the text (lines 122 and 124).

You used CA-AKI and CI-AKI (lines 199, 201, 204, 243, 253). Maybe it is better to uniform this through the text.

Is there any explanation for increase lactate level (7.9 mol/L), two weeks after catheterization and 7 days after CRRT stopped (Figure 2).

Author Response

Dear Reviewer,

We appreciate that you took the time to review our manuscript.

This case report is related to contrast associated acute kidney injury requiring CRRT in a neonate with aortic stenosis. This is very extensive and detailed report based on authors own experience with very good discussion and clear recommandation in conclusion.

I have only some technical comments.

Mark Figure 1 in the text.

We have marked Figure 1 in the text.

Abbreviation KDGIO and CRRT should be named fully when used for the first time in the text (lines 122 and 124).

We spelled the abbreviations after the first using.

You used CA-AKI and CI-AKI (lines 199, 201, 204, 243, 253). Maybe it is better to uniform this through the text.

We modified accordingly

Is there any explanation for increase lactate level (7.9 mol/L), two weeks after catheterization and 7 days after CRRT stopped (Figure 2).

The increase in lactate level was due to a failed extubating attempt two weeks after catheterization, leading to heart and respiratory failure exacerbation.

Kind regards,

The authors

Reviewer 4 Report

The presented case report describes the possibility of the occurrence of AKI due to contract injection in infants. The presented case report is a valuable addition to a paper published in 2013 in Critical Care Journal. 

I do not have any major comments, since the case report is fairly written, all the procedures are described in detail as well as the discussion is well written. 

I would only suggest: 

Please town down little bit your conclusions since this is only a single case report - please also discuss any limitations of your conclusions. 

Is there any data regarding a family history of renal diseases in this particular case? 

What was the ethnicity of the patient? 

Is there any data regarding complex morphology? 

Minor typos and grammatical errors were found. 

Author Response

Dear Reviewer,

We appreciate that you took the time to review our manuscript.

The presented case report describes the possibility of the occurrence of AKI due to contract injection in infants. The presented case report is a valuable addition to a paper published in 2013 in Critical Care Journal. 

I do not have any major comments, since the case report is fairly written, all the procedures are described in detail as well as the discussion is well written. 

I would only suggest: 

Please town down little bit your conclusions since this is only a single case report - please also discuss any limitations of your conclusions. 

The major limitation of our conclusions is the single case experience with managing CA-AKI, and particularly regarding the efficacy of CRRT for this indication.

Is there any data regarding a family history of renal diseases in this particular case?

When evaluating the possible etiologies of AKI, we excluded renal malformations or a family history of renal disease.  

What was the ethnicity of the patient? 

The patient was Caucasian.

Is there any data regarding complex morphology? 

The patient had normal renal morphology for age.

Kind regards,

The authors